# The Fate of IgE Epitopes and Coeliac Toxic Motifs during Simulated Gastrointestinal Digestion of Pizza Base

**DOI:** 10.3390/foods11142000

**Published:** 2022-07-06

**Authors:** Matthew E. Daly, Kai Wang, Xiaoyan Pan, Rosa L. Depau, Justin Marsh, Francesco Capozzi, Phil Johnson, Lee A. Gethings, E. N. Clare Mills

**Affiliations:** 1Manchester Institute of Biotechnology, School of Biological Sciences, Manchester Academic Health Sciences Centre, The University of Manchester, Princess Street, Manchester M1 7DN, UK; matthew.daly-3@postgrad.manchester.ac.uk (M.E.D.); kai.wang-3@postgrad.manchester.ac.uk (K.W.); xiaoyan@tcypherbio.com (X.P.); rosalauradepau@yahoo.it (R.L.D.); justin.marsh@unl.edu (J.M.); philip.johnson@unl.edu (P.J.); lee.gethings@manchester.ac.uk (L.A.G.); 2Department of Food Sciences, University of Bologna, 40126 Bologna, Italy; francesco.capozzi@unibo.it

**Keywords:** wheat, gluten, soy, allergen, coeliac toxic motif, IgE epitope, in vitro digestion, mass spectrometry, allergenicity risk assessment

## Abstract

Understanding how food processing may modify allergen bioaccessibility and the evolution of immunologically active peptides in the gastrointestinal tract is essential if knowledge-based approaches to reducing the allergenicity of food are to be realised. A soy-enriched wheat-based pizza base was subjected to in vitro oral–gastro–duodenal digestion and resulting digests analysed using a combination of sodium dodecyl sulphate-polyacrylamide gel electrophoresis (SDS-PAGE) and mass spectrometry (MS). The digestion profile of pizza base resembled that of bread crust where higher temperatures during baking reduced protein solubility but still resulted in the generation of a complex mixture of peptides. MS profiling showed numerous peptides carrying IgE epitopes, and coeliac toxic motifs were in excess of 20–30 residues long and were only released after either 120 min of gastric digestion or a combination of gastric and duodenal digestion. In silico prediction tools showed an overestimated number of cleavage sites identified experimentally, with low levels of atypical peptic and chymotryptic cleavage sites identified particularly at glutamine residues. These data suggest that such alternative pepsin cleavage sites may play a role in digestion of glutamine-rich cereal foods. They also contribute to efforts to provide benchmarks for mapping in vitro digestion products of novel proteins which form part of the allergenicity risk assessment.

## 1. Introduction

Immune-mediated adverse reactions to food include two types of condition: (1) those known as immediate hypersensitivities where reactions take place within minutes of consumption of a problem food and involve the generation of food-specific IgE, and (2) cell-mediated reactions which manifest over several hours, such as coeliac disease (CD). IgE-mediated food allergies have been estimated to affect up to 5.0% of the population in Europe [1,2], the majority being caused by eight main allergenic foods [3] with the prevalence of probable IgE-mediated wheat and soybean allergy ranging from 0 to 0.37% [1,2]. Allergic reactions can involve the skin, respiratory, gastrointestinal and nervous systems and (rarely) cause severe life threating anaphylactic reactions which can be fatal. Co-factors, such as exercise and non-steroidal anti-inflammatory drugs, may also potentiate food allergic reactions such as exercise being associated with IgE-mediated reactions to wheat in a condition known as wheat-dependent exercise-induced anaphylaxis (WDEIA) [4]. In comparison, CD is estimated to affect 0.5%–1% of the worldwide general population, predominately women, with many remaining undiagnosed [5]. The main clinical manifestation of CD involves a T-cell-mediated inflammatory reaction which results in flattening of the mucosa in the small bowel, a phenomenon largely confined to the duodenum [6], which in turn results in malabsorption of nutrients.

Despite such differences, the molecular triggers that cause both IgE-mediated wheat allergies and CD are proteins. Major wheat (*Triticum aestivum*) allergens belong to seed storage prolamins and include low molecular weight glutenin subunit (LMW-GS) (Tri a 36) [7], a γ-gliadin (Tri a 20) [8] and an ω5-gliadin (Tri a 19) [9]. The major soybean (*Glycine max*) allergens include the 7S and 11S seed storage globulins known as Gly m 5 and Gly m 6, respectively [10], and the 2S albumin allergen Gly m 8 [11]. Other soybean allergens include the Bet v 1 homologue Gly m 4, sensitisation to which is associated with severe reactions to certain types of processed soy ingredients [12]. In contrast, CD is triggered by ingestion of gluten from cereals wheat, barley, rye and oats and consequently certain gluten proteins, such as Tri a 36, Tri a 20 and Tri a 19, can cause both IgE-mediated allergies and CD.

The sites within proteins responsible for these types of immune-mediated adverse reactions are dictated by the underlying cellular pathology. Thus, allergens must contain multiple IgE binding sites (also known as epitopes) in order to cross-link IgE bound to basophils or mast cells. This results in the release of pre-formed mediators which in turn cause the physiological changes that manifest themselves as an allergic reaction. In CD nine amino acid residue T-cell epitopes derived from gluten, found in digestion resistant gluten protein fragments and known as coeliac toxic motifs, proteins bind to the human leukocyte antigen (HLA) DQ receptor (HLA-DQ) on antigen presenting cells in susceptible individuals. These cells subsequently present the peptide sequence to gluten reactive CD4^+^ T cells, triggering pro-inflammatory cytokine release and causing the symptoms associated with CD [13]. Coeliac toxic motifs are widely distributed across the different types of seed storage prolamins [14] with the number of coeliac toxic motifs in a gluten protein being correlated to its immunotoxicity.

Bioaccessibility of food proteins in the gastrointestinal tract is thought to play a key role in determining the form in which they are presented to the gut mucosal immune system to cause either IgE-mediated food allergies or CD. Studies on purified allergens have shown that several wheat (e.g., Tri a 14 [15]) and soy allergens (e.g., Gly m 5 [16]) are resistant to digestion. However, the ω5-gliadin allergen, Tri a 19, appears to be more susceptible to digestion although the resulting peptides retain sufficient structural integrity to still bind IgE [17]. Similarly, the 33mer peptide (LQLQPFPQPQLPYPQPQLPYPQPQLPYPQPQPF) derived from α2-gliadin, one of the most well-characterised immunogenic and immunodominant coeliac toxic peptides to be identified, is derived from a simulated digest of purified α2-gliadin [18]. However, food processing and the food matrix may affect allergen bioaccessibility and resistance to digestion as a consequence of processing-induced unfolding, aggregation and interactions with other food components [19,20,21]. The form of food processing can dramatically alter susceptibility of proteins to digestion with gluten protein digestibility being reduced in baked foods such as bread and muffin compared to cookies [22,23,24,25]. In contrast, a significant proportion of the globulin allergens from the α-amylase/trypsin inhibitor family are released into pasta cooking water, while the majority of pasta proteins except Tri a 19 are digested [26].

The CHANCE project developed several food formulations, including a modified pizza base, aimed at delivering high quality dietary protein at an affordable price [27]. We have now investigated the effect of this modified baked matrix on the bioaccessibility of IgE epitopes and coeliac toxic motifs from gluten and soybean proteins following simulated gastro-duodenal digestion. Digestion models mimicking oral-gastric digestion followed by duodenal digestion were used since they reflect digestion conditions in the region of the small intestine where significant flattening of the mucosa is observed [6]. This digestion has also previously been applied to analysis of the digestion of other cereal foods [22,23,24,25]. Protein digestion was monitored using a combination of electrophoresis and untargeted mass spectrometry (MS).

## 2. Materials and Methods

### 2.1. Materials

Chemicals were of analytical grade and purchased either from Sigma-Aldrich (Dorset, UK) or Fisher Scientific UK Limited (Hertfordshire, UK) unless otherwise stated. Enzymes were purchased as freeze-dried powder as previously described [27]. NuPAGE^®^ LDS sample buffer, 4%–12% NuPAGE^®^ pre-cast Novex^®^, NuPAGE^®^ MES buffer gel, Simply Blue Stain and See Blue plus 2 marker were obtained from Invitrogen, Thermo Fisher Scientific (Hertfordshire, UK). Pizza base fortified with soya paste was prepared as previously described [28] using the following ingredients pre-warmed to room temperature: 88 g wheat flour, 60 g water, 20 g soya yaso product, 5 g wheat fibre, 4 g brewer’s yeast, 4 g olive oil, 2 g sugar and 1 g salt. Ingredients were mixed in a household mixer (BIMBY robot–Vorwerk) for 5 min with inversion of rotation direction every 30 s. The dough was leavened in a thermoclimatic chamber at 30 °C with 90% relative humidity for 2 h. The leavened dough was subsequently baked for 12 min at 250 °C.

### 2.2. Methods

#### 2.2.1. Profiling of Wheat and Soybean Allergens’ Pfam Domain

UniProt accession numbers for WHO/IUIS database allergens were recovered and corresponding sequences downloaded from UniProt (accessed 26 February 2021) in FASTA format [29]. The Pfam domain listed for each accession was collated and the Pfam clan that family belonged to was also retrieved [30]. The number of distinct Pfam domains/clans of all wheat and soy allergens was analysed using GraphPad Prism 8. Protein accessions are available in Appendix A.

#### 2.2.2. Batch Digestion Model

Simulated digestion was carried out in three phases: oral, gastric and duodenal. Briefly, the pizza base was treated with a mincer and simulated salivary fluid (SSF, 0.15 M NaCl, 6 μg/mL lysozyme, 0.2 U human salivary amylase/mg carbohydrate, pH 7) was added at a 3:1 (*w*:*v*) ratio to simulate chewing. The pizza base oral digest was mixed for a further 5 min with shaking at 170 rpm at 37 °C. A subsample of this mix was taken for further analysis, and the remainder subjected to gastric digestion where this sample was mixed with simulated gastric fluid (SGF, 0.9 mM NaH_2_PO_4_, 3 mM CaCl_2_, 0.1 M HCl, 0.15 M NaCl, 16 mM KCl, 183 U porcine pepsin/mg protein, 1.4 U lipase from *Aspergillus niger*/mg fat, pH 2.5) at a 1:1 (*v*:*v*) ratio. The pH was then adjusted to 2.5 using 1 M HCl, and the mix incubated at 37 °C with shaking at 170 rpm. Pepsin was inactivated at 0.3, 60 and 120 min, respectively, by addition of 0.5 M NaHCO_3_ to raise the pH above 7.5 and checked with pH paper [27]. The gastric control sample (GU) contained no digestive enzymes but underwent the same pH transitions and incubations as samples containing gastric enzymes. Gastric digesta stopped at 0.3 and 60 min underwent simulated duodenal digestion. These samples were mixed with hepatic mix solution (HMS, 0.68 mL, 12.5 mM sodium taurocholate, 12.5 mM sodium glycodeoxycholate, 146 mM NaCl, 2.6 mM CaCl_2_, 4.8 mM KCl, 4 mM cholesterol, pH 6.5) for 10 min at 37 °C with shaking prior to addition of pancreatic mix solution (PMS, 0.6 mM CaCl_2_, 4.1 μM ZnSO_4_, 125 mM NaCl, 0.3 mM MgCl_2_, 34.5 U porcine trypsin/mg protein, 0.4 U bovine α-chymotrypsin/mg protein, 1.7 U porcine α-amylase/mg carbohydrate and 1 U porcine lipase/mg fat). The mixing ratio of gastric digesta, PMS and HMS is 1:0.5:0.17 (*v*:*v*:*v*). After incubation with shaking, digestion by chymotrypsin and trypsin was quenched at 0.3, 60 and 120 min by addition of phenylmethylsulfonyl fluoride to 10 mM. Digesta were subsequently stored at −20 °C for further analysis.

#### 2.2.3. Gel Electrophoresis

Digested samples were thawed and centrifuged at 13,200× *g* for 5 min at 4 °C and the soluble fraction collected. Protein content was determined using a BCA assay in a 96 well plate format (Sigma-Aldrich, UK) using bovine serum albumin as the standard protein. Supernatants were mixed 1:1 (*v*/*v*) with NuPAGE^®^ LDS sample buffer containing 10 mM DTT and heated at 90 °C for 10 min. Twenty micrograms of protein were loaded per lane (4%-12% NuPAGE^®^ pre-cast Novex^®^ gel) and the buffer chamber was filled with NuPAGE^®^ MES buffer. Samples were separated for 40 min at a constant voltage 200 V. Gels were fixed in 40% (*v*/*v*) methanol and 10% (*w*/*v*) trichloroacetic acid for 90 min, washed for 2 × 5 min using 100 mL distilled water prior to staining for 5 h using Simply Blue Safe Stain (Invitrogen, UK). The gel was subsequently imaged using a GE Healthcare Typhoon TRIO variable mode imager (GE Healthcare Lifesciences, Buckinghamshire, UK).

#### 2.2.4. Protein Mass Spectrometry and Data Analysis

The soluble fractions of digested samples were analysed by reverse-phase liquid chromatography electrospray ionisation mass spectrometry (RP-LC-ESI-MS) using a Nanoacquity nano-LC (Waters, Manchester, UK) coupled to an Orbitrap Elite mass spectrometer (Thermo Scientific, Waltham, MA, USA) to obtain mass spectra in the positive mode. Samples were separated based on hydrophobicity using a 1.7 µm ACQUITY™ C18 75 μm × 100 mm (Waters, Manchester, UK) analytical column prior to injection. Peptides were eluted from the column with an initial isocratic elution at 2% (*v*/*v*) acetonitrile:water for 10 min, before applying a linear gradient up to 60% (*v*/*v*) acetonitrile:water over 25 min. This was followed by a linear increase to 90% (*v*/*v*) acetonitrile:water at 50 min. Formic acid was included at 0.1% (*v*/*v*) in both solvents throughout the run. The mass range for the survey scans was *m*/*z* 300–2000, resolution 60,000, with *m*/*z* values determined by the Orbitrap Fourier transform mass spectrometry (FTMS) stage. The FTMS fill target was 700,000 ions with a maximum fill time of 1000 ms. The resultant monoisotopic masses were accurate to less than 5 ppm. MS/MS spectra were obtained using collision induced dissociation with collision voltage 35 V and *m*/*z* values determined by the Linear Ion Trap stage. MS/MS was triggered by a minimal signal of 5000 ions with a fill target of 30,000 ions and 150 ms maximum fill time. A maximum of 5 MS/MS spectra per survey scan were obtained by defaulting to the most abundant ions, with *m*/*z* values determined to be less than ~0.4 Da. Charge state selection was not enabled. Dynamic exclusion was set to 1 count and 180 s exclusion with an exclusion mass window of −0.5 to +1.5 Da.

Mass spectral data were searched against a combined database containing sequences attributed to *Triticum aestivum* (n = 139,547) and *Glycine max* (n = 85,034) downloaded from UniProt (accessed on 2 February 2021) using PEAKS Client 6.0 build 2012620 (Bioinformatic Solutions Inc., Waterloo, ON, Canada) with Bioworks v. 3.3.1 (Thermo Scientific, Waltham, MA, USA) installed. The digestion enzyme was set to none, with carbidomethylation of cysteine set as a fixed modification and oxidation of methionine, hydroxylation of proline and deamidation of glutamine or asparagine set to variable modification. Parent ion mass error tolerance was 10 ppm and fragment mass tolerance was 0.5 Da. The PEAKS score was set to −10lgP ≥ 10 for peptide and ≥20 for protein, with the protein false discovery rate controlled at ≤1%. Proteins were only considered identified if at least one unique peptide could be found within the data. The molecular function of identified proteins was determined using the gene ontologies tool QuickGO (https://www.ebi.ac.uk/QuickGO; accessed 21 November 2021). The entire collection of gene ontology annotations was filtered using UniProt accessions of identified proteins and the aspect was set to “molecular function” with assignment limited to InterPro. The statistics page was exported and resulting data plotted in GraphPad Prism 8 for Windows (GraphPad Software, San Diego, CA, USA). The MS data have been deposited in the PRoteomics IDEntifications (PRIDE) Archive database [31], with a dataset identifier of PXD031067.

#### 2.2.5. In Silico Digestion and Peptide Mapping

PeptideCutter (https://web.expasy.org/compute_pi/; accessed 11 January 2022) was used to predict gastrointestinal endoprotease cleavage sites using a specificity model for pepsin pH > 2, a low specificity model for chymotrypsin (C-term to [FYWML], not before P) and the default model for trypsin. Peptides were mapped onto the parent protein sequences and the number of spectral counts of each individual amino acid within that sequence calculated using the PeptideExtractor script (https://github.com/eparker05/PeptideExtractor/tree/master/PepEx; accessed 20 October 2021) and Python 2.7 (Python Software Foundation. Python Language Reference, version 2.7. Available at http://www.python.org; accessed 20 October 2021). The output values from this script were then plotted using GraphPad Prism 8. Visualisation of peptides mapped onto the parent protein sequence was achieved using MATLAB R202a (MathWorks, Natick, MA, USA) and a custom script named Pepdraw v4 (Andrew Watson, Institute of Food Research, Norwich, UK). For spectral counting, peptide sequences present at more than one location within the parent protein were removed as the specific location of the peptide within the sequence could not be determined.

#### 2.2.6. IgE-Binding Epitopes and Coeliac Toxic Motif Analysis

Sequences of IgE-binding epitopes present in proteins of wheat and soybean were retrieved from previously published articles [32,33,34,35]. Coeliac toxic motifs were retrieved from the AllergenOnline Celiac Disease (CD) Novel Protein Risk Assessment Tool (accessed 6 October 2021, (http://www.allergenonline.org/celiachome.shtml; accessed 27 October 2021) [36]. Peptides identified from MS database searching were interrogated to determine if they contained any IgE epitopes or coeliac toxic motifs.

## 3. Results

### 3.1. Pfam Domain Analysis of Curated Wheat and Soy Allergen Sequences

In order to support MS analysis of digests, curated databases of wheat (*Triticum aestivum*; Aller-wheat) and soybean (*Glycine max*; Aller-soy) allergen sequences were assembled based on UniProt accessions in the WHO/IUIS allergen nomenclature database (Appendix A). The criteria for inclusion in the WHO/IUIS database are based on (1) presence in the source, (2) molecular characterisation and (3) proof of specific recognition by at least five sera of human donors [37]. However, further curation was required to allow effective interpretation of annotations to be made, as some details, such as exposure route, were not always supported by the literature cited. For example, Tri a 43, a theoretical wheat protein of unknown function, is listed in IUIS with a primary route of exposure listed as “food”, although the supporting publication detailing this protein focuses on allergens associated with an occupational inhalant allergy known as bakers’ asthma [38]. Therefore, the primary route of exposure should be listed as “airway” rather than “food”. The soy allergen Gly m 2, a defensin protein, did not relate to a specific allergen isoform and consequently was not included in the database. Following curation, 34 wheat and 14 soy allergen sequences were identified and collated in FASTA format (available at 10.6084/m9.figshare.17306804 and 10.6084/m9.figshare.17306822; Appendix A).

The curated allergen sequences were analysed with respect to the Pfam domain present within that protein [30] (Appendix A) and organised by Pfam clan (Figure 1). Overall, wheat allergens represented a much more diverse range of proteins, spanning 20 different Pfam families, while soy allergens were only classified into 5 families. As expected, due to the large proportion of seed storage prolamins within wheat grain, the prolamin clan (CL0482) was dominated by wheat allergens. It includes members of the gliadin family (PF13016; the γ-gliadin allergen Tri a 20 and the α-gliadin Tri a 21) and the protease inhibitor/seed storage/LTP family (PF00234; includes the α-amylase/trypsin inhibitor allergens Tri a 15, 28, 29, 30 and 40, the nsLTP allergen Tri a 14 and the soybean 2S albumin allergen Gly m 8). Another soy allergen, the hydrophobic seed protein family (PF14547; Gly m 1), also belongs to CL0482. The cupin clan (CL0029) was the second most represented but was dominated by the soybean allergens belonging to the Pfam cupin_5 family PF00190 with seven accessions attributed to Gly m 5 (β-conglycinin, the 7S seed storage globulin) and Gly m 6 (glycinin, the 11S globulin). Cupins, such as triticin and germins, are also found in wheat seeds, and there is some evidence to suggest wheat germins are allergenic [39]. The profilin-like clan (CL0431) included the wheat (Tri a 12) and soybean (Gly m 3) allergens which both belong to the profilin family (PF00235). The remaining allergens were grouped as “other”, including the Pfam clan thioredoxin, Bet v 1-like clan (soybean Gly m 4) and six other clans. Tri a 26 (HMW-GS) is located within the Glutenin_HMW Pfam family but is not assigned to a clan. Several IUIS allergens have not been assigned to a protein family, including the allergen Tri a 19 (ω5-gliadin) and Gly m 7.

### 3.2. Gel Electrophoresis of Simulated Digests of Pizza Base

Initially, the bioaccessibility and susceptibility of pizza base proteins to in vitro oral-gastrointestinal digestion was monitored by gel electrophoresis of the soluble fraction of digests (Figure 2). As observed previously for bread, the gluten proteins were relatively poorly stained, especially in the gastric digests (Figure 2A). The protein profile of the “oral digest” (Figure 2A) showed only two major bands of Mr ~55 kDa that most likely correspond to the salivary amylase present in the SSF. Some poorly resolved faintly staining lower molecular weight protein was also evident together with a band of Mr ~14 kDa likely to correspond to lysozyme. A similar protein profile could also be observed in the GU sample (without pepsin gastric control) although it was more faintly stained (Figure 2A). As expected, the 0.3-min gastric digest (G0) was dominated by the band corresponding to pepsin (Mr ~40 kDa), which was accompanied by the appearance of a faint Mr ~22 kDa band together with faintly staining polypeptides of similar Mr to those seen in the oral digest. After 60 min of gastric digestion (G60), the bands corresponding to salivary amylase and the 22 kDa polypeptide had disappeared while the putative lysozyme band was still visible, indicating it was resistant to pepsin digestion. These were accompanied by unresolved bands of Mr < 12 kDa corresponding to digestion products. The 120 min (G120) time point demonstrated a similar electrophoretic profile to that of the G60 sample, suggesting that G60 corresponded to the endpoint of digestion.

Consequently, the G0 and G60 gastric digests were selected to undergo simulated duodenal digestion (Figure 2B,C). Enzymes added at each stage were visible as major bands including pancreatic α-amylase (Mr ~54 kDa), pepsin (Mr ~38 kDa), chymotrypsin and trypsin (Mrs ~28 and 25 kDa, respectively) and lysozyme (Mr ~14 kDa) (Figure 2B). Additional faintly staining bands were also visible at Mrs ~36.5, 30, 20, 21, 15, 12 and 10 kDa in the 0.3- and 60-min digests (D0 and D60, respectively) which likely correspond to intact and partially degraded pizza proteins together with degradation products of the digestive enzymes. After 120-min duodenal digestion (D120), the protein profile was similar to that of the D60 sample, being dominated by digestive enzymes with the putative pepsin band having been completely digested. Other bands of Mr ~54, 40 and 38 kDa became more prominent while two bands of Mr ~21 and ~19 kDa disappeared. Duodenal digestion of the G60 sample only showed evidence of digestive enzymes (Figure 2C) with the lack of staining between 14 and 22 kDa reflecting more extensive gastric digestion in this sample.

### 3.3. Peptide Profiling of Simulated Digests of Pizza Base by Mass Spectrometry

Peptide digestion products in the soluble fractions of all gastric samples and duodenal digests of the G60 sample were profiled by MS (Figure 3). Profiling was limited to peptides >5 amino acids in length due to the MS acquisition mode and search parameters used. A relatively small number of wheat and soy peptides were identified after 0.3 min of simulated gastric digestion (G0) which increased as digestion proceeded with a total of 3532 wheat- and 474 soy-derived peptides being identified after 120 min (Figure 3A,C). During duodenal digestion of the G60 sample, the number of wheat peptides increased after only 0.3 min (D0 sample) and only decreased after 120 min duodenal digestion. In contrast, peptides attributed to soy decreased in the D0 sample compared to the G60 sample but then remained largely stable for the remainder of the duodenal digestion timecourse (Appendix A).

Although some very long peptides of up to 38 amino acid residues in length were identified, the median lengths ranged from 10 to 15 residues (Figure 4B,D). Peptide length was significantly shorter in the D120 compared to the G0 sample for both wheat (*p* < 0.0001) and soy (*p* = 0.0015). However, the pattern of digestion differed between the two food ingredients. Thus, the peptide length distributions for wheat were similar across time points G0, G60, G120 and D0, with a median length of 14–15 amino acids which decreased to 12 following more extensive duodenal digestion. In contrast, the median peptide length for soy was shorter, especially following duodenal digestion where the median soy peptide length was reduced to 10 in the D60 and D120 samples.

### 3.4. Protein Profiling of Simulated Digests of Pizza Base by Mass Spectrometry

The peptide profiling data was then used to identify the proteins from which the peptides originated (Figure 4 and Appendix A). Some protein identifications appeared and disappeared during digestion (Appendix A). This was likely due to transient generation and subsequent degradation of unique identifying peptides. At each time point, more wheat proteins were identified compared to soy, likely due in part to availability of protein sequences on UniProt (wheat = 139,547 and soya = 85,034) and the large proportion of wheat flour compared to soy paste in the pizza base. The number of wheat proteins identified increased from G0 (109 proteins) to G120 (316 proteins) where it was maximal (Figure 4A), decreasing during subsequent duodenal digestion at D0 and D60 (192 and 122, respectively) before increasing again to 142 at D120. Conversely, the number of soy proteins identified peaked at G60 with 86 after a relatively small number of protein identifications at timepoint G0 (21). Soy protein identifications fell consistently from G60 to D60, where only 27 proteins were identified, before increasing again to 43 proteins at D120.

Protein identifications were also analysed by their gene ontology (GO) and grouped by molecular function (Appendix A). A greater diversity of protein function, as indicated by GO terms, was found for wheat in the gastric compared to duodenal digests (Appendix A). The majority of proteins were attributed to nutrient reservoir activity consistent with the fact that total flour protein comprises ~80% seed storage prolamins [40]. The second GO term with most representations was serine-type endopeptidase inhibitor activity, reflecting the abundance of members of the α-amylase/trypsin inhibitor family in wheat grain. Molecular function analysis of soy proteins revealed more diverse functionality with some GO terms found consistently throughout both gastric and duodenal digestion, including nutrient reservoir activity, oxidoreductases and metal ion binding (Appendix A). The G120 sample provided the most complex GO term profile with four GO terms representing half of the total number of protein identifications, the remainder being split between twelve other GO terms.

An analysis of allergens identified in the gastro-duodenal digests was also undertaken using the curated allergen sequence databases Aller-wheat and Aller-soy (Figure 4B and Appendix A). Overall, the number of isoforms detected increased during gastric digestion and then gradually reduced during duodenal digestion (Appendix A). Specific allergens identified included the seed storage prolamin allergens, ω5-gliadin allergen (Tri a 19 identified in the D120 sample), the HMW glutenin allergen Tri a 26 (identified in all but the D0 digest) and the LMW subunit of glutenin allergen Tri a 36 (identified only in the D0 and D60 digests). Four allergens belonging to the α-amylase/trypsin inhibitor family were also identified in gastric and/or duodenal digests (Tri a 28, Tri a 29, Tri a 30 [CM3] and Tri a 40 [CM17]). Lastly, two metabolic enzyme allergens (triose phosphate isomerase [Tri a 31] and glyceraldehyde-3-phosphate dehydrogenase [Tri a 34]) and α-purothionin (Tri a 37) were identified only in gastric digests. No unique or non-unique peptides were identified which corresponded to the UniProt accession attributed directly to Tri a 14, D2T2K2. However, other nsLTP isoforms, homologous with D2T2K2, were identified in the duodenal digestion D120 sample, such as P24296. The number of allergen isoforms identified at the protein level for soy was found to increase from none to six between the G0 and G60 timepoints, the same number being identified in the G120 digest and belonging entirely to either the 7S (Gly m 5 including the β, α- and α’- subunits) or 11 S (Gly m 6 corresponding to the G1, 2 and 3 isoforms, respectively) seed storage globulins. Although most were still identified in the D0 digests, only three isoforms remained, two of which corresponded to the α- and α’-subunits of the 7S globulin.

### 3.5. Bioaccessible Epitopes and Coeliac Toxic Motifs

An analysis of bioaccessible IgE-binding epitopes (for wheat and soy) and coeliac toxic motifs (wheat only) was then undertaken using IgE epitopes retrieved from the literature [32,33,34,35] and coeliac toxic motifs retrieved from Allergen Online [14]. Only one soy peptide carrying an IgE epitope was identified in the G60 sample, two in the G120 sample and one in the D120 sample (data not shown). A greater number of IgE epitopes than coeliac toxic motifs were identified in the wheat-derived peptides in all time points, both types of peptide increasing throughout gastric digestion (Figure 5A). This number increased further during subsequent duodenal digestion, reaching a maximum in the D60 sample after which it began to decrease. A total of 55 peptides which contained at least two IgE epitopes were identified in the G120 sample and 180 in the D120 sample.

Since the immunological activity of peptides is affected by their length this was also taken into account in the analysis (Figure 5). Peptides carrying IgE epitopes were largely generated following duodenal digestion with a small number of longer peptides carrying IgE epitopes identified in the gastric digestion and the 0.3-min duodenal digest of the G60 time point (D0, Figure 5A,B) with a broad peak of transient peptides 21–28 amino acids in length. This was reflected by an increase in median peptide length from 8–10 residues in gastric digests to around 15 amino acids in the early gastro-duodenal digest, which subsequently decreased in the D120 digest. Although less frequent, peptides containing coeliac toxic motifs tended to be much longer (Figure 5C). After 120 min of gastric digestion, an increased frequency of peptides 13–16 amino acids long was observed which was accompanied by a second peak of peptides 24–30 amino acids long, with nine peptides of 36 residues in length being identified. Duodenal digestion of the G60 sample generated a significant number of additional peptides carrying coeliac toxic motifs, with a broad peak of peptides 21–26 residues in length which were digested after 60 min.

### 3.6. Peptide Mapping of a LMW-GS Allergen Tri a 36

In order to illustrate the patterns of digestion in more detail, the peptides corresponding to wheat allergen Tri a 36 (UniProt accession B2Y2Q7) were mapped onto the protein sequence and tracked throughout the simulated gastro-duodenal digestion (Figure 6). This protein was chosen since it has been identified as an allergen in IgE-mediated wheat allergy [7] and contains several coeliac toxic motifs. Peptides used for mapping included both those unique to isoform B2Y2Q7 (which were only identified in timepoints D0 and D60) and other non-unique peptides also present in other isoforms.

Few peptides were located in the N-terminal portion (residue ~20–100) of the protein in the gastric digestion samples, consistent with the fact that only six pepsin cleavage sites were predicted in this region, confirming its pepsin resistant nature (Figure 6A, Appendix A). Peptides corresponding to the repetitive sequences occurred at multiple positions within the repetitive domain of Tri a 36 (sequence 34–197) and consequently could not be unambiguously mapped onto the LMW-GS subunit sequence (blue shaded areas of sequence, Figure 6, Appendix A). For example, the peptide QQQPPFSQQQPPFSQQ, which spans the IgE epitopes “QQQPP”, was generated by pepsin, identified in the G60 timepoint and is found at positions 77–92 and 115–130. A larger ensemble of nested and overlapping peptides spanning the repetitive region was identified in the duodenal digests which also comprised both coeliac toxic motifs and at least one IgE epitope “QQQPP”. Removed repetitive sequences had a combined spectral count of 31 in D0, 46 in D60 and 59 in D120 (Supplementary Information S3). The majority of peptides identified lay within the central and C-terminal regions and were also nested and overlapping with a number spanning both coeliac toxic motifs and IgE epitopes (Figure 6A). There was also evidence of ragged processing of the C-terminus of the protein which probably originated in the seed and has been observed previously for LMW subunits of gluten with loss of C-terminal tyrosine [41]. Although individually of low abundance, collectively, the peptides increased in abundance in the G120 sample compared to the G60 sample showing slow but progressive digestion, particularly of the central region of the protein (Figure 7A).

The gastro-duodenal digests showed more extensive proteolysis, with 164 peptides identified in the D0 sample which gave a sequence coverage of 85%, the most complete coverage of any timepoint (Figure 6B). Four regions were not covered since they would give rise to peptides < 5 amino acids in length, either directly (residues 230–234 and 315–320) or as a consequence of tryptic cleavage (residues 261–268 and 334–344). Such short peptides would not be readily detected by the MS methods employed in this study (Figure 6B). Twenty-eight peptides were identified in the D0 sample which arose from gastric digest peptic cleavage and were also present in the preceeding G60 sample. Peptides corresponding to the N-terminal region were identified which included several coeliac toxic motifs (‘FSQQQQQPL’, ‘QQPPFSQQQQPPFSQ’, ‘QQPPFSQQQQPVLPQ’, ‘PFSQQQQPV’) and arose as a consequence of chymotrypsin activity.

A much larger number of cleavage sites were predicted for all the endoproteases than were identified experimentally (Table 1). Furthermore, some cleavage sites were predicted to be in common for pepsin and chymotrypsin, many of which were experimentally verified in the gastro-duodenal digests. Missed cleavages for chymotrypsin were only observed at Phe and Leu, possibly indicating other factors, such as local secondary structure, may affect cleavage by this enzyme. In addition, more than half of the cleavage sites identified experimentally in the gastric digests were not consistent with generally accepted pepsin cleavage rules. For example, while pepsin preferentially cleaves at either the N- or C-terminal of Phe, Tyr, Trp and Leu at pH >2 [43], many peptides identified in the C-terminal region of the protein in both G60 and G120 were derived from two other cleavage sites, Thr^354^-Thr^355^ and Met^344^-Cys^345^. Analysis of the experimentally determined cleavage sites identified that a significant proportion were located at either the N- or C-terminal side of a Gln residue (Appendix A). Comparison of these atypical cleavage sites in the simulated gastric and gastro-duodenal digests showed that they were generated by both pepsin and chymotrypsin; however, the majority were of low relative abundance (Figure 6). The exception was a peptide generated by pepsin, ^1^^42^ SQQQLPPFSQQQSPFSQQQQ ^161^, which was moderately abundant in both the G60 and G120 samples but lost in the gastro-duodenal samples.

As gastro-duodenal digestion proceeded, sequence coverage was reduced to 59% in the D120 sample reflecting the fact that further digestion resulted in peptides too small to be determined by LC MS-MS (Figure 6C and Figure 7B and Appendix A). For example, one region at position 228–239 at timepoint G60 is partially covered by seven peptides. The most abundant peptides identified in the D120 sample lay in the N-terminal region (Figure 6C) with six peptides identified containing coeliac toxic motifs including ‘PFSQQQQPV’, ‘FSQQQQQPL’ and the IgE epitope ‘QQQPP’. A total of nine peptides were identified consistently in each sample apart from G0, indicating they were highly resistant to both gastric and duodenal digestion. These peptides contained no predicted tryptic cleavage sites; however, some did contain chymotryptic cleavage sites. Further, eight out of nine of these resistant peptides originated between position 190 and 227, suggesting that the structure of this region may contribute to its resistant nature.

## 4. Discussion

The SDS-PAGE protein profile of the chew and digest samples showed only a limited solubilisation of pizza base proteins which is similar to previously published digestion profiles of bread crust [20,21] and is significantly less complex than the profile seen for previous studies of bread crumb digestion [20,21,22,44]. Bread crust protein solubility during digestion is significantly reduced compared to that of bread crumb, likely due to more extensive protein modification in the crust which reaches temperatures of 180–200 °C as compared to ~100 °C for bread crumb [20,21].

While pizza base proteins appeared less soluble, the digests proved to have a rich repertoire of peptides when profiled using MS. Although several allergen isoforms were only transiently identified, many wheat and soybean allergens were consistently identified from the in vitro digests. This may be due to the way in which proteolysis generated unique peptides, some of which contained missed cleavage sites which were subsequently digested. This further complicated annotation of gluten proteins where generation of unique peptides for unequivocal identification is troublesome due to their homologous nature [14]. Investigation of the linked UniProt accessions within the WHO/IUIS nomenclature database revealed some mis-annotations. For example, Tri a 12 (profilin) is listed as a food allergen within the database. However, profilin proteins are typically found in pollen and as such would likely not pose a risk to someone with food allergy to wheat and were not identified in the pizza base analysis presented here. Furthermore, protein level evidence of some of the allergens listed by the WHO/IUIS database is lacking and the quality of related evidence listed in UniProt is low. Therefore, allergen sequence curation is required prior to undertaking MS proteomic analysis.

In silico tools increasingly play a crucial role in modern scientific research. However, the predictions made using in silico digestion analysis performed poorly, overestimating the number of peptic, tryptic and chymotryptic cleavages sites. Furthermore, novel cleavage sites were observed for pepsin and chymotrypsin, albeit that they were only hydrolysed to a limited extent. The data presented in this report highlight the difficulties in predicting proteolysis in complex mixtures of proteins in processed food matrices rather than peptide substrates or purified proteins and the need to take this into account to support analysis of complex mixtures, such as those found in food digests, in future studies.

The vast majority of peptides identified in the digests which contained either coeliac toxic motifs or IgE epitopes were more than nine amino acids in length. Indeed, many longer transient peptides, including peptides over 20 amino acids in length, were generated by either 60- or 120-min gastric digestion followed by 0.3-min duodenal digestion. These findings are consistent with those of Ogilvie, et al. [44] who showed that immunogenic peptides are released from bread within the first 60 min of simulated gastric digestion. However, only one marker peptide (RPQQPYPQPQPQY) used by Ogilvie, et al. [44] was detected in this study at timepoint D0. This is likely due to differences between the studies regarding wheat cultivars used [14], baking protocol or in vitro digestion method. It has been proposed that peptides longer than nine amino acids in length have the potential to act as T-cell epitopes, triggering coeliac disease [45], with peptides of between nine and twenty-three residues in length being able to act as substrates for tissue transglutaminase [46]. IgE-mediated allergic reactions require activation of T cells but also involve development of humoral (i.e., antibody) responses requiring the activation of B cells; B cell epitopes require a length of at least 15 amino acids to trigger activation [47]. Peptides carrying coeliac toxic motifs were found to persist and, although some carried IgE epitopes which might have the potential to stimulate antibody responses, they were too small and did not carry the multiple IgE epitopes required to activate effector cells and cause an allergic reaction.

## 5. Conclusions

The bioaccessibility of pizza base proteins resembled that of proteins from bread crust which have been exposed to higher temperatures during baking than the crumb of conventional bread loaves. Those proteins that were released into solution were rapidly digested using an in vitro oral–gastro–duodenal digestion system to give a diverse and complex mixture of peptides. The peptide profiles evolved over time, with long digestion time courses required in excess of 60 min to release wheat gluten peptides carrying coeliac toxic motifs. This illustrates the synergistic effect of a gastro-duodenal digestion system, rather than simply relying on gastric digestion alone, to investigate generation of large resistant peptides which carry coeliac toxic motifs and multiple IgE epitopes. Such mapping undertaken in well-characterised allergenic foods, such as wheat and soybean, can support interpretation and similar characterisation of in vitro digestion products from newly expressed proteins which forms part of allergenicity risk assessment [48,49]. Data sets, such as the ones generated in this study, will support development of new approaches to predicting proteolysis in future studies. However, as with any proteomic analysis, the MS data analysis undertaken to identify allergenic proteins, peptides, coeliac toxic motifs and IgE epitopes relies on having access to well-curated allergen sequences. This study further emphasises the need for a wider consensus as to what constitutes an allergen and evidence-based approaches to compiling curated allergen sequences to realise the potential of proteomic analysis in understanding the impact of food processing on protein digestion as supported by Selb et al. [50].

## Figures and Tables

**Figure 1 foods-11-02000-f001:**
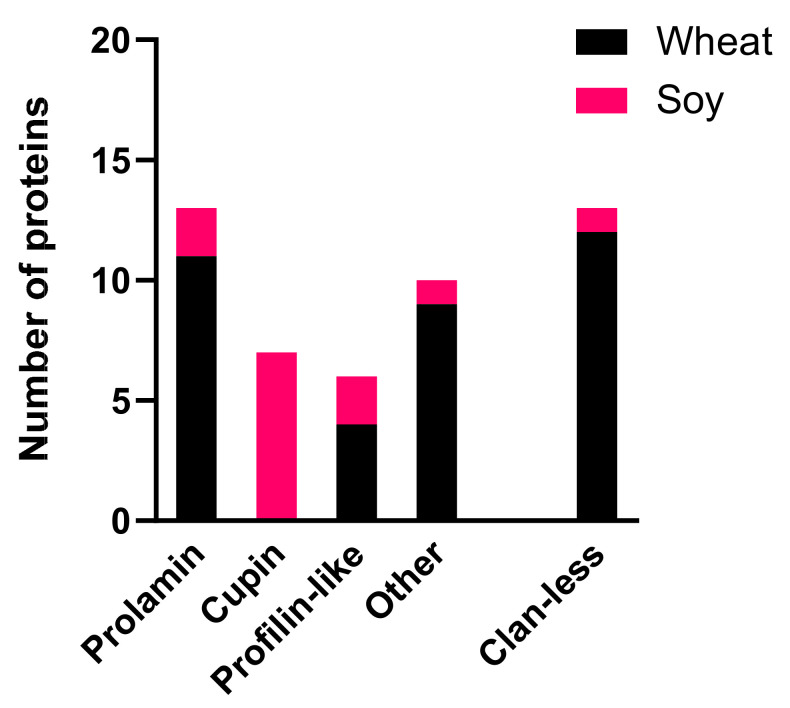
Wheat (black) and soy (red) allergens present in the WHO/IUIS allergen nomenclature database grouped according to Pfam clan [30]. Clans in the “other” category include CI-2, Glyco_hydro_tim, TIM_barrel, GADPH_aa-bio_dh, NADP_Rossmann, Thioredoxin, Zn_Beta_Ribbon and Bet_v_1_like.

**Figure 2 foods-11-02000-f002:**
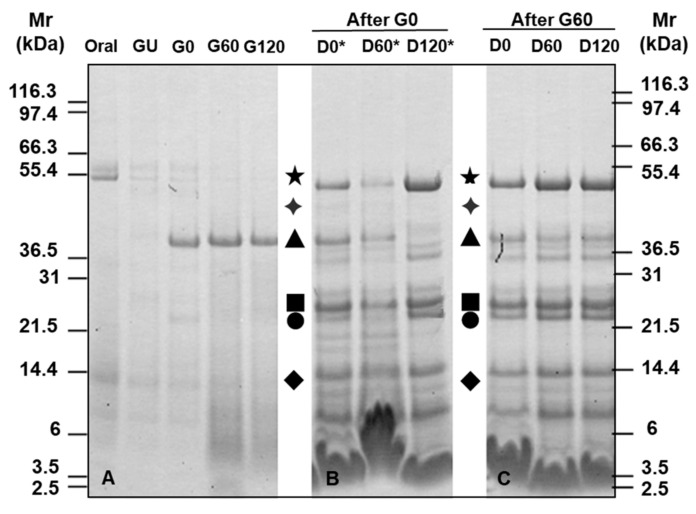
SDS-PAGE profiles of the soluble fraction of in vitro oral, gastric and duodenal digests of pizza base. (**A**) Oral and gastric digestion stopped at the following digestion time points: ‘GU’- no pepsin 0.3-min control, ‘G0’—0.3 min, ‘G60’—60 min, ‘G120’—120 min. (**B**) G0 digests subjected to duodenal digestion stopped at duodenal digestion time points as follows: ‘D0’—0.3 min, ‘D60’—60 min, ‘D120’—120 min. (**C**) G60 digests subjected to duodenal digestion; timepoints are as in (B). Symbols indicate bands corresponding to enzymes included in the in vitro simulated digestion as follows: salivary α-amylase (★); pepsin (▲); pancreatic α-amylase (**✦**); chymotrypsin (■); trypsin (●); lysozyme (♦). Timepoints suffixed with * indicate duodenal digestion following 0.3 minutes of simulated gastric digestion.

**Figure 3 foods-11-02000-f003:**
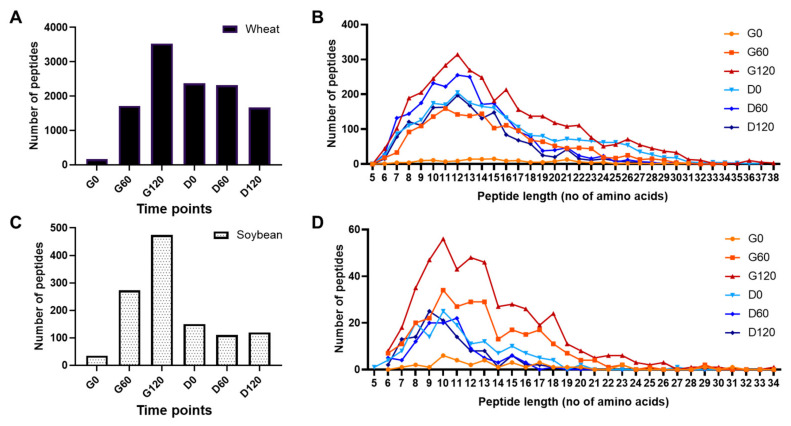
Peptide profiles of pizza base digests defined by LC-MS/MS analysis. Number of identified wheat (**A**) and soy peptides (**C**). Peptide length distributions for wheat (**B**) and soy (**D**) peptides. Digestion time points are for gastric (G) and duodenal (D) digests after 0.3, 60 and 120 min.

**Figure 4 foods-11-02000-f004:**
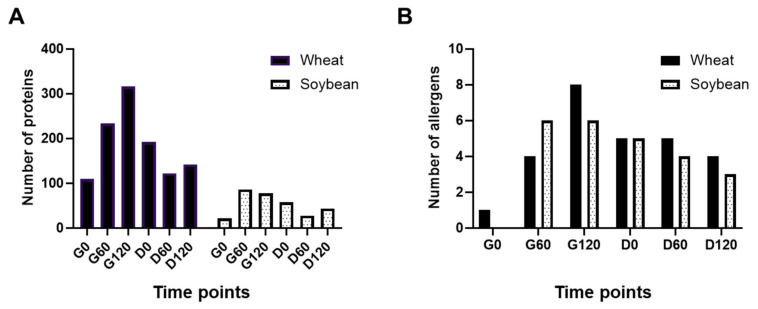
Identification of wheat and soy proteins by mass spectrometry analysis of the soluble fraction of in vitro gastric (G) and duodenal (D) digestion samples. Filled bars—wheat; dotted bars—soy. (**A**) Total number of proteins identified by PEAKS with at least one unique peptide and FDR ≤ 1%. (**B**) Allergen isoforms were identified using the Aller-wheat and Aller-soy sequence databases. Digestion time points are as described in Figure 2C.

**Figure 5 foods-11-02000-f005:**
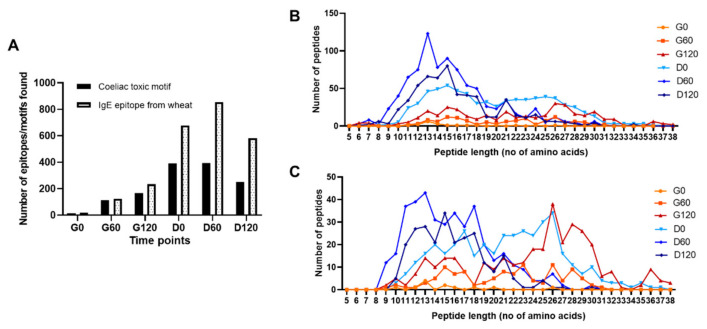
Bioaccessible wheat-derived peptides carrying IgE epitopes and coeliac toxic motifs in pizza base digests. (**A**) The number of non-unique IgE epitopes and coeliac toxic motifs found within identified peptides attributed to wheat throughout simulated digestion. (**B**,**C**) Length distribution of wheat peptides found containing non-unique IgE epitopes (**B**) and coeliac toxic motifs (**C**) throughout simulated digestion. Digestion time points are as described in Figure 2.

**Figure 6 foods-11-02000-f006:**
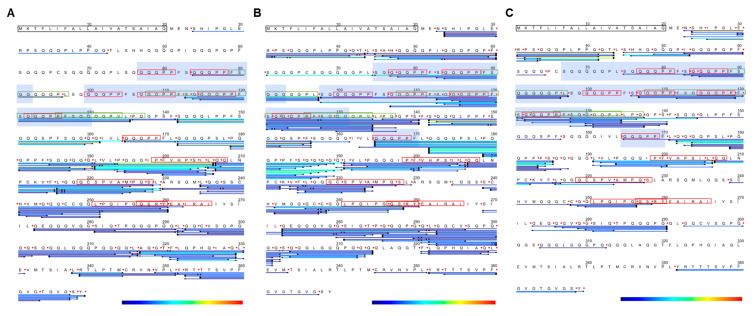
Peptide mapping of the LMW-GS allergen B2Y2Q7 throughout simulated digestion. Identified peptides were mapped onto the parent protein B2Y2Q7 at timepoints G60 (**A**), D0 (**B**) and D120 (**C**). The sequences bounded by a black box indicate the signal sequence, and those bounded in red and green represent IgE epitopes and coeliac toxic motifs, respectively. The domains with consensus repetitive peptides are shown in blue boxes. Endoprotease cleavage sites are denoted by red dots. Only the G60 digest is shown since it was similar to the G120 digest (Appendix A). Colour shading represents relative abundance of peptides calculated using spectral counting with the gradient from blue to red indicating increasing abundance.

**Figure 7 foods-11-02000-f007:**
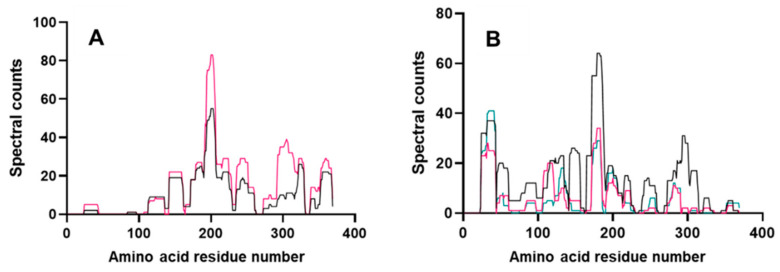
Semi-quantitative analysis of peptide coverage of LMW GS B2Y2Q7 following simulated gastric (**A**) and gastro-duodenal digestion (**B**). Spectral counts [42] were plotted as a function of amino acid residue as follows: panel (**A**)—spectral counts of G60 labelled in black and G120 in red; panel (**B**)—D0 labelled in black, D60 in red and D120 in green.

**Table 1 foods-11-02000-t001:** Predicted and experimentally determined cleavage sites for the LMW-GS allergen B2Y2Q7 throughout simulated digestion. ‘n.a.’ not applicable. The specificity models used for prediction were set as follows: pepsin—pH > 2; chymotrypsin—low specificity; trypsin—default.

	No. of Cleavage Sites					
	Pepsin	Chymotrypsin	Pepsin/Chymotrypsin	Trypsin	Alternative	Total
**Predicted**	47	49	21	7	n.a.	82
**Experimentally Determined**						
G60	32	n.a.	n.a.	n.a.	53	85
G120	38	n.a.	n.a.	n.a.	70	108
D0	28	35	15	2	79	144
D60	23	31	14	2	66	122
D120	17	25	10	0	70	112

## Data Availability

The mass spectrometry proteomics data have been deposited to the ProteomeXchange Consortium via the PRIDE partner repository with the dataset identifier PXD031067.

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
