# Peer review of "The Fate of IgE Epitopes and Coeliac Toxic Motifs during Simulated Gastrointestinal Digestion of Pizza Base"

_foods, 2022, doi:10.3390/foods11142000_

Round 1

Reviewer 1 Report

Reviewer comments:

In this manuscript (foods-1666702), the authors investigated how baking influence on the fate of the wheat and soy allergens present in the soy-enriched wheat-based pizza base during simulated in vitro digestion. It is well known, that wheat and soybean belong to eight main allergenic food, which can caused IgE-mediated allergy or coeliac disease (only wheat). Beside this, it is known that food processing may modify allergens susceptibility to digestion and their allergenicity, so understanding of these processes is useful for lowering allergenicity of the food products. In this manuscript, the authors first defined all wheat and soy allergens and their isoforms using WHO/IUIS and UniProt databases and their belonging to the specific Pfam clan. In the next step, they analyzed the number and length of peptides obtained during in vitro oral-gastric-intestinal digestion, as well as which proteins they belonging. Which of these proteins are allergens, and how many peptides which bearing IgE-binding epitopes and coeliac toxic motifs were generated by these allergens? In the last step, they used the wheat allergen Tri a 36 to illustrate the patterns of digestion in more detail, and showed differences in the number and type of cleavage sites by used peptidases using in silico and in vitro approach.

This manuscript is very interesting and valuable for readers, because it reveals that pizza base proteins are less soluble than breadcrumb proteins. This soluble fraction contains the significant number of wheat and soy allergens, which can give numerous peptides during the simulated gastrointestinal digestion.  The peptides carrying IgE epitopes are commonly longer than peptides containing coeliac toxic motifs and they were found in all digestion time points. In this form, the manuscript needs only minor corrections, which are listed below before the final decision. Please, take into account below comments and suggestions for the improvement of your article's quality.

Keywords

Use italic for in vitro

Abstract

Currently, the abstract has 216 words. According to the instruction for the authors, it should have up to 200 words.

Line 18 and 23

Replace duodenal with intestinal

Introduction

Lines 38, 40, 46 and in other parts of the manuscript

Please, insert space between number and %.

Line 40

Replace 0-0.37 with 0 to 0.037

Line 47-50

The end of sentence about flattening of the gut mucosa should be checked or changed. The process of digestion starts in the duodenum, while jejunum and ileum are the main sites for the absorption of nutrients, so if they are damage, malabsorption occurs.

Line 51

Please, insert and soybean after wheat.

Line 52, and 54

Please, insert Latin name of plants after wheat (Triticum aestivum) and soybean (Glycine max)

Line 53

Put Tri a 36 in parentheses.

Line 54

Insert the comma before respectively.

Line 85

Please replace the sentence: Other soybean allergens include the Bet v 1 homologue, Glee m 4, … [12]., with Other soybean allergens include the Gly m 4, a member of the superfamily of the Bet v 1 homologous proteins, sensitisation….[12].

Line 61

Insert reactions after adverse.

Line 64

Replace and with or.

Line 67

Replace the human leukocyte antigens (HLA) HLA-DQ receptor with the human leukocyte antigens DQ  (HLA-DQ) receptor.

Lines 70-72

Replace, Coeliac toxic motifs widely distributed across the different types of seed storage prolamins [14], the number of coeliac toxic motifs in a gluten protein being correlated to its immunotoxicity, with Coeliac toxic motifs are widely distributed across the different types of prolamins, the class of seed storage gluten proteins [14]. It is known, that the number of coeliac toxic motifs present in a structure of specific gluten protein being correlated to its immunotoxicity (Insert refrence 14, if this conclusion is from this reference).

Line 77

Replace the ω5‐gliadin allergen, Tri a 19, with Tri a 19

Line 86

Insert of proteins after susceptibility.

Line 87

Insert comma after in contrast

Line 89

Insert while before the majority

Materials and Methods

Lines 97, 110, 111, 141, 153, 192

Delete space and/or full stop before title.

Line 106

Insert space in and1.

Lines 108, 109… and in other parts of the manuscript

Insert space before units for example ℃, M, W, µm, cm, V, Da, s  or %

Lines 119 and 132

Replace duodenal with intestinal.

Line 121 and 133 and in other parts of the manuscript

Use mL instead ml.

Line 125

Insert which volume of SGF was added.

Line 158

Please, check the column characteristics 75 μm x 10 cm

Line 126

Use italic for Aspergillus niger.

Line 175

Delete (wheat) and (soybean).

Line 190

Replace full stop with comma.

Lines 191

Use italic for In silico.

Line 208-210

Replace the sentence: Sequences of IgE-binding epitopes present in wheat proteins were retrieved from Matsuo, Yokooji [32], whereas those present in soy were retrieved from Saeed, Gagnon [33], He and Xi [34] and Kern, Havenith [35].  With Sequences of IgE-binding epitopes present in proteins of wheat and soybean were retrieved from previously published articles [32-35].

Results

Lines 217 and 218

Delete (Triticum aestivum; Aller-wheat) and (Glycine max; Aller-soy)

Lines 219

Insert (Supplementary Table S1) after database.

Line 244

Insert Gly m before 6.

Line 248

Replace clan with clans.

Line 256

Delete full stop.

Lines 258 and 281

Italic in vitro.

Lines 261 and 262

Delete of the “chew” and add after oral digestion

Line 263

Replace Some poorly resolved lower molecular weight protein staining was evident together with a band of Mr ~ 14 kDa likely to correspond to lysozyme. Beside amylase, some poorly resolved and stained LMW proteins are present together with a band of Mr ~ 14 kDa likely to correspond to lysozyme.

Line 265

Replace no with without.

Line 266

Insert comma after pepsin.

Line 269

Replace chew with oral digest.

Figure 2

Please replace Chew with Oral and letter D with letter I

Lines 276-281

SDS-PAGE profiles of the soluble fraction of in vitro oral, gastric and duodenal digests of pizza base. (A) Chew and gastric digestion, stopped at the following digestion timepoints: ‘GU’- no pepsin 0.3 min control, ‘G0’- 0.3 mins, ‘G60’- 60 min, ‘G120’- 120 min. (B) G0 digests subjected to duodenal digestion, stopped at duodenal digestion time points as follows: ‘D0’-  0.3 mins, ‘D60’- 60 min, ‘D120’- 120 min. (C) G60 digests subjected to duodenal digestion; timepoints are as in (B).

Replace with

SDS-PAGE profiles of the soluble fraction of in vitro oral, gastric (G) and intestinal (I) digests of pizza base. (A) Oral and gastric digests, gastric digestion stopped at the following time points: 0.3, 60 and 120 min (G0, G30 and G120, respectively). GU- without pepsin, gastric control at time point 0.3 min, (B) G0  and (C) G60 digests subjected to intestinal digestion, stopped at the following time points 0.3, 60 and 120 min (I0, I60 and I120 respectively).

Lines 285, 294, 299, 305, 308, 320, 321 and in other parts of manuscript

Replace duodenal with intestinal.

Line 287

Delete kDa after 28.

Line 288

Insert at time points I0 and I60 before faintly staining

Line 289

Delete in the 0.3 and 60 min digests (D0 and D60 respectively).

Line 291

Replace duodenal digestion (D60) with of intestinal digestion (I120) of G0 digest,

Line 292

Replace D60 with I60

Line 302

Replace after 0.3 min of simulated gastric digestion (G0) with at time point G0,

Line 303

Insert gastric between as and digestion.

Lines 302-304

A relatively small number of wheat and soy peptides were identified after 0.3 min of simulated gastric digestion (G0), which increased as digestion proceeded with a total of 3532 wheat- and 474 soy-derived peptides being identified after 120 min. This sentence replace with the next sentence:

A relatively small number of wheat and soy peptides were identified at time point G0, which increased as gastric digestion proceeded, and reached the total of 3532 wheat- and 474 soy-derived peptides after 120 min of digestion (G120).

Line 304

There is a discrepancy in the total number of wheat peptides in G120 in the main text, Figure 3, and Table S2. Please check this data.

Line 306

(D0 sample) only decreasing after 120 min duodenal digestion, replace with (I0), then decreased at I60 and I120, and reached the number lower than in G60.

Line 307

Replace D0 with I0.

Lines 311-313

Peptide profiles of pizza base digests defined by LC-MS/MS analysis. (A, C) Number of identified wheat (A) and soy peptides (C); Peptide length distributions for wheat (B) and soy (D) peptides. Digestion time points are as described in Fig.2.

Peptide profiles of in vitro gastric (G) and intestinal (I) of digestion of pizza base defined by LC-MS/MS analysis. Number of identified wheat (A) and soy (C) peptides; Peptide length distributions for wheat (B) and soy (D) peptides. Digestion time points are 0.3, 60 and 120 min (0, 60 and 120, respectively).

Figure 3

Replace letter D with I.

Line 314

Insert space into upto.

Lines 316, 319 and in other parts of manuscript

Please, replace D0, D60 and D120 with I0, I60 and I120

Line 322

Replace in the D60 and D120 samples with at time points I60 and I120.

Lines 328

Insert space in timepoint

Figure 4.

Please, replace D with I

Line 340

Insert gastric (G) and intestinal (I) after in vitro.

Line 342

Insert by PEAKS after identified and inset UniProt after using. Delete Aller before wheat and soy.

After are insert 0.3, 60 and 120 min (0, 60, 120, respectively). Delete as described in Fig 2.

Lines 358

Insert wheat and soy after curated and delete Allert-whet and Allert-soy.

Lines 360-363

Replace

Specific allergens identified included the seed storage prolamin allergens, ω5-gliadin allergen (Tri a 19; identified in the D120 sample), the HMW glutenin allergen Tri a 26 (identified in all but the D0 digest), and the LMW subunit of glutenin allergen Tri a 36 (identified only in 363 the D0 and D60 digests).

with

Specific allergens identified included the seed storage prolamin allergens, Tri a 19 (I120), Tri a 26 (in all time points except I0), and Tri a 36 (I0 and I60).

Lines 364 - 367

Replace

Four allergens belonging to the α-amylase/trypsin inhibitor family were also identified in gastric and/or duodenal digests (Tri a 28, Tri a 29, Tri a 30 [CM3] and Tri a 40 [CM17]). Lastly, two metabolic enzyme allergens (triose phosphate isomerase [Tri a 31] and glyceraldehyde-3-phosphate dehydrogenase [Tri a 34]) and α-purothionin (Tri a 37) were identified only in gastric digests.

With

Four allergens (Tri a 28, Tri a 29, Tri a 30 and Tri a 40) belonging to the α-amylase/trypsin inhibitor family were also identified in gastric and/or intestinal digests. Lastly, two metabolic enzyme allergens (Tri a 31 and Tri a 34) and Tri a 37 were identified only in gastric digests.

Lines 369 - 371

Replace Uniprot with UniProt. Insert isoform after D2T2K2. Delete nsLTP. Replace in the duodenal digestion D120 sample with at time points I120.

Line 371-375

Replace

…  timepoints, the same number being identified in the G120 digest and belonged entirely to either the 7S (Gly m 5; including the β, α- and α'- subunits) or 11 S (Gly m 6; corresponding to the G1, 2 and 3 isoforms respectively) seed storage globulins.

with

… time points, the same number being identified in the G120 digest and belonged entirely to either Gly m 5 or Gly m 6 allergen, which belong to seed storage globulins.

Line 377

Replace α- and α'- subunits of the 7S globulin with Gly m 5 allergen.

Line 378

Insert IgE-binding before epitopes

Line 381

Insert [32-35] after literature

Line 384

Insert in all time points after peptides.

Figure 5

Line 392

Insert (gastic (G) and intestinal (I)) after digests

Line 395

Describe time points as in Figure 3 or 4.

Line 399

Delete the 0.3min duodenal digest of the G60 399 time point (D0, and insert at time points I0 of intestinal digestion.

Lines 401-403

Replace 8-10 with 11 to 18. Replace early gastroduodenal digest with I0 time point. Replace D120 digest with I120 time point.

Line 413 and in other parts of manuscript

Replace gastroduodenal with gastrointestinal.

Line 441

Insert gastrointestinal before digestion.

Line 429

Replace removed with Removed

Line 451

Replace D with I.

Table 1

In the title of table, add gastrointestinal before digestion. Replace D with I in table.

Lines 471 and 479

Replace phenylalanine and leucine with Phe and Leu, and glutamine with Gln.

Discussion

Line 518

Replace p with P in UniProt

References

Please, check the order of references from 41 to 45 in the main text. The references 42 and 45 did not cited in the main text.

Supplementary material

Table S1.

Replace,  The Allerwheat and Allersoy allergen database lists  with Pfam domain analysis of wheat (Triticum aestrvum) and soy (Glycine max) allergens listed in the WHO/IUIS database.

Table S2.

Replace  timepoints throughout simulated digestion., with time points (0.3, 60 and 120 min,  0, 60 and 120, respectively) throughout simulated gastric (G) and intestinal (I) digestions.

Please, in table S2 replace letter D with I, and insert space in timepoints. Also, check the total number of wheat proteins at time point G120. It should be 3532, this value is in the main text and figure 3.

Table S3 and S4

Wheat allergens identified in each digestion timepoint with at least one unique peptide. Highlighted orange box indicates that UniProt accession was found in that timepoint.

Please, insert space in timepoint.

Add by PEAKS after identified.

Replace letter D with I in Table S3 and S4.

Supplementary information S1 and S2

Replace letter D with I in these tables.

Figure S1 and S2

Replace D with I in both titles.

Please in Figure S2 insert specificity of each oxidoreductase activity on panel B.

Explain why this analysis yields more protein than the PEAKS analysis at the same time points.

Figure S3

Insert throughout simulated gastrointestinal digestion

Author Response

Manuscript ID: foods-1666702

Daly et al  “The fate of IgE epitopes and coeliac toxic motifs during simulated gastro-intestinal digestion of pizza base”.

Response to Reviewers Comments

Reviewer 1

Reviewer comments:

In this manuscript (foods-1666702), the authors investigated how baking influence on the fate of the wheat and soy allergens present in the soy-enriched wheat-based pizza base during simulated in vitro digestion. It is well known, that wheat and soybean belong to eight main allergenic food, which can caused IgE-mediated allergy or coeliac disease (only wheat). Beside this, it is known that food processing may modify allergens susceptibility to digestion and their allergenicity, so understanding of these processes is useful for lowering allergenicity of the food products. In this manuscript, the authors first defined all wheat and soy allergens and their isoforms using WHO/IUIS and UniProt databases and their belonging to the specific Pfam clan. In the next step, they analyzed the number and length of peptides obtained during in vitro oral-gastric-intestinal digestion, as well as which proteins they belonging. Which of these proteins are allergens, and how many peptides which bearing IgE-binding epitopes and coeliac toxic motifs were generated by these allergens? In the last step, they used the wheat allergen Tri a 36 to illustrate the patterns of digestion in more detail, and showed differences in the number and type of cleavage sites by used peptidases using in silico and in vitro approach.

This manuscript is very interesting and valuable for readers, because it reveals that pizza base proteins are less soluble than breadcrumb proteins. This soluble fraction contains the significant number of wheat and soy allergens, which can give numerous peptides during the simulated gastrointestinal digestion.  The peptides carrying IgE epitopes are commonly longer than peptides containing coeliac toxic motifs and they were found in all digestion time points. In this form, the manuscript needs only minor corrections, which are listed below before the final decision. Please, take into account below comments and suggestions for the improvement of your article's quality.

Keywords

Use italic for in vitro

Response: The text format has been modified.

Abstract

Currently, the abstract has 216 words. According to the instruction for the authors, it should have up to 200 words.

Response: According to the word count in word it was 201 words in the submitted version but it maybe that this has a different mechanisms for counting words. The text has been revised so the abstract is now 200 words long, as per the word count function in Word.

Line 18 and 23

Replace duodenal with intestinal

Response: The digestion model we used employed a pH of 6.5 in order to simulate the conditions found in the duodenum where the pH is lower than the remainder of the intestine due to efflux of very low pH gastric contents. This was considered more relevant for this study since the effect of coeliac disease is more severe in the duodenum. Further along the intestinal tract, in regions such as the jejunum and ileum, the pH increases to at least 7.8. Hence we have used the term “duodenal digestion” to reflect these differences. There are other models which seek to replicate conditions further down the intestinal tract and employ a higher pH value and might be more correctly referred to, for example, as jejunal models. Therefore, the models are referred to throughout as duodenal to highlight this distinction. The text has been modified at the end of the introduction (p3, lines 96-98) in order to explain this.

Introduction

Lines 38, 40, 46 and in other parts of the manuscript

Please, insert space between number and %.

Response: The formatting has been corrected as indicated by the reviewer.

Line 40

Replace 0-0.37 with 0 to 0.037

Response: The formatting has been corrected as indicated by the reviewer.

Line 47-50

The end of sentence about flattening of the gut mucosa should be checked or changed. The process of digestion starts in the duodenum, while jejunum and ileum are the main sites for the absorption of nutrients, so if they are damage, malabsorption occurs.

Response: In coeliac disease the intestinal mucosa is flattened from the duodenum which is particularly severe, with lesser flattening in the ileum (Freeman, “Pearls and pitfalls in the diagnosis of adult celiac disease.” Canadian journal of gastroenterology 22 (2008): 273-80. doi:10.1155/2008/905325). The text in the introduction has been modified to further clarify this.

Line 51

Please, insert and soybean after wheat.

Response: The major soybean allergens are detailed from line 53 and the sentence indicated by the reviewer refers only to the gluten proteins so the text as been left unaltered.  

Line 52, and 54

Please, insert Latin name of plants after wheat (Triticum aestivum) and soybean (Glycine max)

Response: The Latin names have been added at lines 51 and 54.

Line 53

Put Tri a 36 in parentheses.

Response: The brackets have been added.

Line 54

Insert the comma before respectively.

Response: The punctuation has been modified.

Line 85

Please replace the sentence: Other soybean allergens include the Bet v 1 homologue, Glee m 4, … [12]., with Other soybean allergens include the Gly m 4, a member of the superfamily of the Bet v 1 homologous proteins, sensitisation….[12].

Response: The typographical error has been corrected but otherwise the original sentence construction has been retained as the proposed change modified the meaning.

Line 61

Insert reactions after adverse.

Response: The missing word has been added.

Line 64

Replace and with or.

Response: The grammatical error has been corrected.

Line 67

Replace the human leukocyte antigens (HLA) HLA-DQ receptor with the human leukocyte antigens DQ  (HLA-DQ) receptor.

Response: The sentence has been reworded.

Lines 70-72

Replace, Coeliac toxic motifs widely distributed across the different types of seed storage prolamins [14], the number of coeliac toxic motifs in a gluten protein being correlated to its immunotoxicity, with Coeliac toxic motifs are widely distributed across the different types of prolamins, the class of seed storage gluten proteins [14]. It is known, that the number of coeliac toxic motifs present in a structure of specific gluten protein being correlated to its immunotoxicity (Insert refrence 14, if this conclusion is from this reference).

Response: The missing word from the original sentence has been added to ensure it reads correctly.

Line 77

Replace the ω5‐gliadin allergen, Tri a 19, with Tri a 19

Response: The allergen Tri a 19 is the name given to one ω5‐gliadin sequence in IUIS for which IgE binding has been confirmed. There are homologues of that sequence, however their IgE reactivity has not been confirmed. The text as originally written keeps this sense more closely than the change indicated by the reviewer.

Line 86

Insert of proteins after susceptibility.

Response: The missing word has been added.

Line 87

Insert comma after in contrast

Response: The punctuation has been modified.

Line 89

Insert while before the majority

Response: The missing word has been added.

Materials and Methods

Lines 97, 110, 111, 141, 153, 192

Delete space and/or full stop before title.

Response: The manuscript has been prepared using the Food word template and consequently the convention used is that provided by the journal and has not been revised.

Line 106

Insert space in and1.

Response: The formatting has been corrected as indicated by the reviewer.

Lines 108, 109… and in other parts of the manuscript

Insert space before units for example ℃, M, W, µm, cm, V, Da, s  or %

Response: The formatting has been modified as indicated by the reviewer.

Lines 119 and 132

Replace duodenal with intestinal.

Response: This has not been modified for the reasons indicated above.

Line 121 and 133 and in other parts of the manuscript

Use mL instead ml.

Response: The formatting has been corrected as indicated by the reviewer.

Line 125

Insert which volume of SGF was added.

Response: The volume was added as a ration 1:1 (v:v) as indicated on line 129. We provide the information in this manner so that anyone wishing to repeat the experiments can do it since it is the ratio of SGF to food that is important.

Line 158

Please, check the column characteristics 75 μm x 10 cm

Response: The column dimensions are correct, however it has been modified to read “75 μm x 100mm”.

Line 126

Use italic for Aspergillus niger.

Response: The latin name has been italicised.

Line 175

Delete (wheat) and (soybean).

Response: The words have been deleted.

Line 190

Replace full stop with comma.

Response: The punctuation has been modified.

Lines 191

Use italic for In silico.

Response: “In silico” is already formatted in italics.

Line 208-210

Replace the sentence: Sequences of IgE-binding epitopes present in wheat proteins were retrieved from Matsuo, Yokooji [32], whereas those present in soy were retrieved from Saeed, Gagnon [33], He and Xi [34] and Kern, Havenith [35].  With Sequences of IgE-binding epitopes present in proteins of wheat and soybean were retrieved from previously published articles [32-35].

Response: The sentence has been modified as suggested by the reviewer.

Results

Lines 217 and 218

Delete (Triticum aestivum; Aller-wheat) and (Glycine max; Aller-soy)

Response: The names in the text refer to sequence databases that have been complied for Triticum aestivum and Glycine max allergen sequences, respectively and so the names Aller-wheat, and Aller-soy have been retained in the manuscript.

Lines 219

Insert (Supplementary Table S1) after database.

Response: The wording has been revised as indicated by the reviewer.

Line 244

Insert Gly m before 6.

Response: The missing name has been added.

Line 248

Replace clan with clans.

Response: The text has not been modified as the wording is correct.

Line 256

Delete full stop.

Response: It was not possible to locate the missing full stop in the manuscript.

Lines 258 and 281

Italic in vitro.

Response: The latin name has been italicised.

Lines 261 and 262

Delete of the “chew” and add after oral digestion

Response: The context and figure 2 have been modified as indicated by the reviewer.

Line 263

Replace Some poorly resolved lower molecular weight protein staining was evident together with a band of Mr ~ 14 kDa likely to correspond to lysozyme. Beside amylase, some poorly resolved and stained LMW proteins are present together with a band of Mr ~ 14 kDa likely to correspond to lysozyme.

Response: The text has been modified to increase clarity although slightly differently from that suggested by the reviewer.

Line 265

Replace no with without.

Response: The wording has been revised.

Line 266

Insert comma after pepsin.

Response: The punctuation has been modified.

Line 269

Replace chew with oral digest.

Response: Wording revised as above.

Figure 2

Please replace Chew with Oral and letter D with letter I

Response: Wording revised as above for chew versus oral digestion but the lettering has been retained since the digestion model used mimics duodenal digestion conditions specifically.

Lines 276-281

SDS-PAGE profiles of the soluble fraction of in vitro oral, gastric and duodenal digests of pizza base. (A) Chew and gastric digestion, stopped at the following digestion timepoints: ‘GU’- no pepsin 0.3 min control, ‘G0’- 0.3 mins, ‘G60’- 60 min, ‘G120’- 120 min. (B) G0 digests subjected to duodenal digestion, stopped at duodenal digestion time points as follows: ‘D0’-  0.3 mins, ‘D60’- 60 min, ‘D120’- 120 min. (C) G60 digests subjected to duodenal digestion; timepoints are as in (B).

Replace with

SDS-PAGE profiles of the soluble fraction of in vitro oral, gastric (G) and intestinal (I) digests of pizza base. (A) Oral and gastric digests, gastric digestion stopped at the following time points: 0.3, 60 and 120 min (G0, G30 and G120, respectively). GU- without pepsin, gastric control at time point 0.3 min, (B) G0 and (C) G60 digests subjected to intestinal digestion, stopped at the following time points 0.3, 60 and 120 min (I0, I60 and I120 respectively).

Response: Wording revised as above for chew versus oral digestion but the wording and lettering has been otherwise retained since the digestion model used mimics duodenal digestion conditions specifically.

Lines 285, 294, 299, 305, 308, 320, 321 and in other parts of manuscript

Replace duodenal with intestinal.

Response: Wording revised as above for chew versus oral digestion but the wording and lettering has been otherwise retained since the digestion model used mimics duodenal digestion conditions specifically.

Line 287

Delete kDa after 28.

Response: The text has been corrected.

Line 288

Insert at time points I0 and I60 before faintly staining

Line 289

Delete in the 0.3 and 60 min digests (D0 and D60 respectively).

Line 291

Replace duodenal digestion (D60) with of intestinal digestion (I120) of G0 digest,

Line 292

Replace D60 with I60

Response: The original lettering and wording has been retained since the digestion model used mimics duodenal digestion conditions specifically.

Line 302

Replace after 0.3 min of simulated gastric digestion (G0) with at time point G0,

Response: The original text has been retained since it reflected more precisely what was done. There is no zero time point – hence G0 is 0.3s the time taken to sample a digestion and some modification does take place even in such a short space of time.

Line 303

Insert gastric between as and digestion.

Response: The text has been modified as indicated by the reviewer.

Lines 302-304

A relatively small number of wheat and soy peptides were identified after 0.3 min of simulated gastric digestion (G0), which increased as digestion proceeded with a total of 3532 wheat- and 474 soy-derived peptides being identified after 120 min. This sentence replace with the next sentence:

A relatively small number of wheat and soy peptides were identified at time point G0, which increased as gastric digestion proceeded, and reached the total of 3532 wheat- and 474 soy-derived peptides after 120 min of digestion (G120).

Response: The original text has been retained – the actual digestion time point is 0.3min not a proper zero time point (c.f. response to the comment above)

Line 304

There is a discrepancy in the total number of wheat peptides in G120 in the main text, Figure 3, and Table S2. Please check this data.

Response: Table S2 has been modified to include the correct value of 3532 wheat peptides at G120. The main text, Figure 3 and Table S2 are now consistent with this.

Line 306

(D0 sample) only decreasing after 120 min duodenal digestion, replace with (I0), then decreased at I60 and I120, and reached the number lower than in G60.

Response: The original text (duodenal digestion, D0, D60, D120) has been retained as per the explanation provided above.

Line 307

Replace D0 with I0.

Response: The original text has been retained as per the explanation provided above.

Lines 311-313

Peptide profiles of pizza base digests defined by LC-MS/MS analysis. (A, C) Number of identified wheat (A) and soy peptides (C); Peptide length distributions for wheat (B) and soy (D) peptides. Digestion time points are as described in Fig.2.

Peptide profiles of in vitro gastric (G) and intestinal (I) of digestion of pizza base defined by LC-MS/MS analysis. Number of identified wheat (A) and soy (C) peptides; Peptide length distributions for wheat (B) and soy (D) peptides. Digestion time points are 0.3, 60 and 120 min (0, 60 and 120, respectively).

Response:  The text has been revised to elaborate the legend and annotation in the figure although slightly differently to the reviewer suggestion. Changes have not been made to the lettering as per the explanation of the digestion models used in the study.

Figure 3

Replace letter D with I.

Response: The original text has been retained as per the explanation provided above.

Line 314

Insert space into upto.

Response: The text has been modified as indicated by the reviewer.

Lines 316, 319 and in other parts of manuscript

Please, replace D0, D60 and D120 with I0, I60 and I120

Response: The original text has been retained as per the explanation provided above.

Line 322

Replace in the D60 and D120 samples with at time points I60 and I120.

Response: The original text has been retained as per the explanation provided above.

Lines 328

Insert space in timepoint

Response: The text has been modified as indicated by the reviewer.

Figure 4.

Please, replace D with I

Response: The original text has been retained as per the explanation provided above.

Line 340

Insert gastric (G) and intestinal (I) after in vitro.

Response: The words inserted “gastric (G) and duodenal (D)” as per the explanation provided above.

Line 342

Insert by PEAKS after identified and inset UniProt after using. Delete Aller before wheat and soy.

After are insert 0.3, 60 and 120 min (0, 60, 120, respectively). Delete as described in Fig 2.

Response: The wording “by PEAKS” has been inserted into the text. Aller-wheat and Aller-soy are the names given to the curated databases detailed earlier in the manuscript (Section 3.1.) containing allergen isoforms present in the WHO/IUIS database, and have been retained in the text.. “As described in Fig 2” has been modified to include “Fig 2 Panel C” to be clear that the sample subjected to duodenal digestion (i.e. timepoint D0) is prepared from a G60 digest.

Lines 358

Insert wheat and soy after curated and delete Allert-whet and Allert-soy.

Response: Aller-wheat and Aller-soy are the names given to the curated databases detailed earlier in the manuscript (Section 3.1.) and have been retained; please see justification above.

Lines 360-363

Replace

Specific allergens identified included the seed storage prolamin allergens, ω5-gliadin allergen (Tri a 19; identified in the D120 sample), the HMW glutenin allergen Tri a 26 (identified in all but the D0 digest), and the LMW subunit of glutenin allergen Tri a 36 (identified only in 363 the D0 and D60 digests).

with

Specific allergens identified included the seed storage prolamin allergens, Tri a 19 (I120), Tri a 26 (in all time points except I0), and Tri a 36 (I0 and I60).

Response: The original text has been retained as per the explanation provided above regarding the digestion systems used. It is more technically correct to include the term digest since it describes the samples prepared under certain digestion conditions and after certain times of digestion.

Lines 364 - 367

Replace

Four allergens belonging to the α-amylase/trypsin inhibitor family were also identified in gastric and/or duodenal digests (Tri a 28, Tri a 29, Tri a 30 [CM3] and Tri a 40 [CM17]). Lastly, two metabolic enzyme allergens (triose phosphate isomerase [Tri a 31] and glyceraldehyde-3-phosphate dehydrogenase [Tri a 34]) and α-purothionin (Tri a 37) were identified only in gastric digests.

With

Four allergens (Tri a 28, Tri a 29, Tri a 30 and Tri a 40) belonging to the α-amylase/trypsin inhibitor family were also identified in gastric and/or intestinal digests. Lastly, two metabolic enzyme allergens (Tri a 31 and Tri a 34) and Tri a 37 were identified only in gastric digests.

Response: The original text has been retained as we feel including the names of the enzymes provides more explanation to those unfamiliar with allergen nomenclature which can be somewhat opaque and difficult to follow.

Lines 369 - 371

Replace Uniprot with UniProt. Insert isoform after D2T2K2. Delete nsLTP. Replace in the duodenal digestion D120 sample with at time points I120.

Response: The typographical error in UniProt has been corrected but the original wording regarding the digests has been retained as per explanations provided above

Line 371-375

Replace

…  timepoints, the same number being identified in the G120 digest and belonged entirely to either the 7S (Gly m 5; including the β, α- and α'- subunits) or 11 S (Gly m 6; corresponding to the G1, 2 and 3 isoforms respectively) seed storage globulins.

with

… time points, the same number being identified in the G120 digest and belonged entirely to either Gly m 5 or Gly m 6 allergen, which belong to seed storage globulins.

Response: The original wording has been retained since it provides greater clarity than simply referring to the allergen names.

Line 377

Replace α- and α'- subunits of the 7S globulin with Gly m 5 allergen.

Response: This is not the sense in which the text is written and we wish to retain the original phrasing.

Line 378

Insert IgE-binding before epitopes

Response: The additional words have been added as indicated by the reviewer.

Line 381

Insert [32-35] after literature

Response: The references have been inserted as requested by the reviewer.

Line 384

Insert in all time points after peptides.

Response: The time points have been added as indicated by the reviewer.

Figure 5

Line 392

Insert (gastic (G) and intestinal (I)) after digests

Line 395

Describe time points as in Figure 3 or 4.

Line 399

Delete the 0.3min duodenal digest of the G60 399 time point (D0, and insert at time points I0 of intestinal digestion.

Lines 401-403

Replace 8-10 with 11 to 18. Replace early gastroduodenal digest with I0 time point. Replace D120 digest with I120 time point.

Line 413 and in other parts of manuscript

Replace gastroduodenal with gastrointestinal.

Line 441

Insert gastrointestinal before digestion.

Response: The original nomenclature has been retained for the reasons stated above.

Line 429

Replace removed with Removed

Response: The text has been modified as indicated by the reviewer.

Line 451

Replace D with I.

Response: The original nomenclature has been retained for the reasons stated above.

Table 1

In the title of table, add gastrointestinal before digestion. Replace D with I in table.

Response: The original nomenclature has been retained for the reasons stated above.

Lines 471 and 479

Replace phenylalanine and leucine with Phe and Leu, and glutamine with Gln.

Response: The amino acids have been abbreviated as indicated by the reviewer.

Discussion

Line 518

Replace p with P in UniProt

Response: UniProt has been correct as indicated by the reviewer.

References

Please, check the order of references from 41 to 45 in the main text. The references 42 and 45 did not cited in the main text.

Response: The references have been checked. Reference 42 was cited in the original manuscript on p12 (now line 474 in the revised manuscript). Reference 45 is  duplicate of reference 22 and has been removed. The remaining references have been renumbered. Reformatting of the references was completed without “track changes” due to the nature of Endnote referencing software.

Supplementary material

Table S1.

Replace,  The Allerwheat and Allersoy allergen database lists  with Pfam domain analysis of wheat (Triticum aestrvum) and soy (Glycine max) allergens listed in the WHO/IUIS database.

Response: The creation of the two allergen databases (Aller-wheat and Aller-Soy, respectively) is detailed in Section 3.1., and Table S1 directly relates to those databases. Gly m 2 has been removed from Table S1 as it did not have a specific UniProt accession and therefore does not appear in the Aller-Soy database. The text has been modified to include “Pfam domain analysis of specific UniProt accessions relating to wheat (Triticum aestivum) and soybean (Glycine max) allergens in the curated Aller-wheat and Aller-Soy databases, respectively.”

Table S2.

Replace  timepoints throughout simulated digestion., with time points (0.3, 60 and 120 min,  0, 60 and 120, respectively) throughout simulated gastric (G) and intestinal (I) digestions.

Please, in table S2 replace letter D with I, and insert space in timepoints. Also, check the total number of wheat proteins at time point G120. It should be 3532, this value is in the main text and figure 3.

Response: The original text has been retained as per the explanation provided above regarding the digestion systems used.

Table S3 and S4

Wheat allergens identified in each digestion timepoint with at least one unique peptide. Highlighted orange box indicates that UniProt accession was found in that timepoint.

Please, insert space in timepoint.

Add by PEAKS after identified.

Response: The text has been modified as indicated by the reviewer.

Replace letter D with I in Table S3 and S4.

Response: The original text has been retained as per the explanation provided above regarding the digestion systems used.

Supplementary information S1 and S2

Replace letter D with I in these tables.

Figure S1 and S2

Replace D with I in both titles.

Response: The original text has been retained as per the explanation provided above regarding the digestion systems used.

Please in Figure S2 insert specificity of each oxidoreductase activity on panel B.

Explain why this analysis yields more protein than the PEAKS analysis at the same time points.

Response: Figure S2 Panel B has been modified to include the specificity of the oxidoreductase. Those annotations labelled only “oxidoreductase” do not have specificity information included in the gene ontology annotation. With regards to the mismatch between the number of protein IDs and GO terms; some protein accessions are annotated with more than one GO term. For example, the UniProt accession A0A0R0F6M9 is annotated with both glucose-1-phosphate adenylyltransferase activity and nucleotidyltransferase activity.

Figure S3

Insert throughout simulated gastrointestinal digestion before at time

Response: The original text has been retained as per the explanation provided above regarding the digestion systems used.

Reviewer 2 Report

The article is well constructed and documented. The main criticism is that there was no immunoreactivity study before and after digestion.

Author Response

Manuscript ID: foods-1666702

Daly et al  “The fate of IgE epitopes and coeliac toxic motifs during simulated gastro-intestinal digestion of pizza base”.

Response to Reviewers Comments

Reviewer 2

The article is well constructed and documented. The main criticism is that there was no immunoreactivity study before and after digestion.

Response: We thank the reviewer for their kind words about the manuscript. We agree that an immunoreactivity study before and after digestion would provide additional value to the work. However, the manuscript is already long and detailed with the mass spectrometry analysis and it was felt such data on immunoreactivity would be better placed in a separate manuscript.  with the reviewer that an analysis.

Reviewer 3 Report

L46: Define “general population.” Worldwide or in the Europe.

L66-67: Rewrite this sentence.

L110-111: Delete the extra period.

L117: Supplementary Table S1 contains several Tabs. Please specify which one is related.

L118: What is the mixing ratio for gastric and pancreatic digestion? Also, are there any references to support this sample preparation method?

L142: Change rpm to g force.

L144-145: How to determine protein concentration accurately after mixing with LDS sample buffer containing DTT?

L148: Please provide accurate information. The gel was run either at a constant voltage or a constant current.

L154: What samples?

L223-225: Rewrite this statement. I was not sure what the authors want to convey.

L226: Which database? Gly m 2 could be searched in WHO/IUIS database (http://allergen.org/viewallergen.php?aid=343).

L230: Again, the authors need to explain each Tab in Supplementary Table S1.

L261: Please explain why they are not stained well at the same loading mass (20 µg).

In Figure 2, why the band intensity for amylase was different in D60 and D120? What is the meaning of *in D0, D60 and D120?

L326: The authors did not mention supplementary Table S2 in this manuscript. Also there is no supplementary Table S4.

Author Response

Manuscript ID: foods-1666702

Daly et al  “The fate of IgE epitopes and coeliac toxic motifs during simulated gastro-intestinal digestion of pizza base”.

Response to Reviewers Comments

Reviewer 3

L46: Define “general population.” Worldwide or in the Europe.

Response: the word “worldwide” has been added in the context on line 46

L66-67: Rewrite this sentence.

Response: The sentence detailing the binding of coeliac toxic motifs to HLA-DQ receptors has been rewritten.

L110-111: Delete the extra period.

Response: The text has been modified as indicated by the reviewer.

L117: Supplementary Table S1 contains several Tabs. Please specify which one is related.

Response: Supplementary Table S1 is present as a single table in the Supplementary Materials file. Supplementary Information S1, S2 and S3 do have several tabs with the tab named as the time point it relates to.

L118: What is the mixing ratio for gastric and pancreatic digestion? Also, are there any references to support this sample preparation method?

Response: This sample preparation method is generally same as Smith et al. 2015 (Molecular Nutrition & Food Research, 59: 2034-43.). The ratios of volume are: Oral digests: Simulated Salivary Fluid is 1: 1 (total = “Gastric digesta”), and Gastric digesta: Pancreatic Mix Solution: Hepatic Mix Solution is 1: 0.5: 0.17. Thus has been specified in the methods section.

L142: Change rpm to g force.

Response: RPM has been changed to the relevant g force.

L144-145: How to determine protein concentration accurately after mixing with LDS sample buffer containing DTT?

Response: The protein content was measured before mixing with LDS buffer and DTT. The sentence order has been changed to clarify this.

L148: Please provide accurate information. The gel was run either at a constant voltage or a constant current.

Response: The gel was run at 200 V with a maximum current of 350 mA. The text has bene modified to include this.

L154: What samples?

Response: The samples were the soluble fraction of digested samples. This has now been  specified in the text.

L223-225: Rewrite this statement. I was not sure what the authors want to convey.

Response: The statement has been rewritten to clarify the meaning of the section.

L226: Which database? Gly m 2 could be searched in WHO/IUIS database (http://allergen.org/viewallergen.php?aid=343).

Response: The reviewer is correct that Gly m 2 is contained within the WHO/IUIS allergen database, however no specific UniProt isoform accession is listed for this allergen and so could not be interrogated regarding identification or Pfam domain analysis.

L230: Again, the authors need to explain each Tab in Supplementary Table S1.

Response: As above, Supplementary Table S1 is present as a single table in the Supplementary Materials file. Supplementary Information S1, S2 and S3 do have several tabs with the tab named as the time point it relates to.

L261: Please explain why they are not stained well at the same loading mass (20 µg).

Response: The major proteins in our samples as gluten proteins which lack basic amino acids (e.g. arginine, lysine and histidine) thus are not effectively stained by Coomassie blue. This has been observed before and is detailed in Smith et al (2015) (Molecular Nutrition & Food Research, 59: 2034-43.).

In Figure 2, why the band intensity for amylase was different in D60 and D120? What is the meaning of *in D0, D60 and D120?

Response: It is likely that the amylase band is gradually digested by the proteases as the time course proceeds. The * means the intestinal digestion was performed using the digests from the G0 sample.

L326: The authors did not mention supplementary Table S2 in this manuscript. Also there is no supplementary Table S4.

Response: Supplementary Table S2 is mentioned in the text in Section 3.3., before Figure 3. Table S4 is mentioned in the text in Section 3.4. Added a further mention of Tables S3 and S4 in Section 3.4.

Round 2

Reviewer 3 Report

The comments were addressed in the revision, and this manuscript could be published.